# Genetic characterization of *Angiostrongylus* larvae and their intermediate host, *Achatina fulica*, in Thailand

**Abdulhakam Dumidae[1], Pichamon Janthu[1], Chanakan Subkrasae[1], Paron Dekumyoy[2], Aunchalee Thanwisai[1,3], Apichat Vitta** [1,3]*

**1** Department of Microbiology and Parasitology, Faculty of Medical Science, Naresuan University, Phitsanulok, Thailand, **2** Department of Helminthology, Faculty of Tropical Medicine, Mahidol University, Ratchavithi Rd, Ratchathewi, Bangkok, Thailand, **3** Centre of Excellence in Medical Biotechnology (CEMB), Faculty of Medical Science, Naresuan University, Phitsanulok, Thailand

* apichatv@nu.ac.th

**Data Availability Statement:** All relevant data are within the manuscript and its Supporting Information files.

## Abstract

Angiostrongyliasis is a parasitic disease caused by nematodes of the genus *Angiostrongylus*. Distribution of this worm corresponds to the dispersal of its main intermediate host, the giant African land snail *Achatina fulica*. Genetic characterization can help identify parasitic pathogens and control the spreading of disease. The present study describes infection of *A. fulica* by *Angiostrongylus*, and provides a genetic outlook based on sequencing of specific regions. We collected 343 land snails from 22 provinces across six regions of Thailand between May 2017 and July 2018. Artificial digestion and Baermann's technique were employed to isolate *Angiostrongylus* larvae. The worm and its intermediate host were identified by sequencing with specific nucleotide regions. Phylogenetic tree was constructed to evaluate the relationship with other isolates. *A. fulica* from Chaiyaphum province was infected with *A. cantonensis*, whereas snails collected from Phrae and Chiang Rai provinces were infected with *A. malaysiensis*. The maximum likelihood tree based on 74 *A. fulica* COI sequences revealed monophyletic groups and identified two haplotypes: AF1 and AF2. Only AF1, which is distributed in all regions of Thailand, harbored the larvae of *A. cantonensis* and *A. malaysiensis*. Two mitochondrial genes (COI and *cytb*) and two nuclear regions (ITS2 and SSU rRNA) were sequenced in 41 *Angiostrongylus* specimens. The COI gene indicated that *A. cantonensis* was closely related to the AC10 haplotype; whereas the *cytb* gene revealed two new haplotypes: AC19 and AC20. SSU rRNA was useful for the identification of *A. cantonensis*; whereas ITS2 was a good genetic marker for differentiating between *A. cantonensis* and *A. malaysiensis*. This study provides genetic information about the parasite *Angiostrongylus* and its snail intermediate host. The data in this work may be useful for further study on the identification of *Angiostrongylus* spp., the genetic relationship between intermediate host and parasite, and control of parasites.

**Funding:** This work was supported by Naresuan University (Grant number R2562B079) to AV.

**Competing interests:** The authors have declared that no competing interests exist.

## Introduction

*Angiostrongylus* is a parasitic nematode from the superfamily Metastrongyloidea [1]. To date, 21 species of this genus have been reported around the world, with *A. cantonensis* and *A. costaricensis* being the most notable [2]. *A. cantonensis* is the causative agent of human angiostrongyliasis associated with eosinophilic meningitis or meningoencephalitis. Ocular and neuroangiostrongyliasis are reported as sporadic in Asian counties [3–5]. *A. costaricensis* causes abdominal angiostrongyliasis and most cases are reported in South America [6,7]. Other veterinary-relevant species include *A. malaysiensis*, *A. mackerrasae*, and *A. vasorum*, which act as animal pathogens [2,8]; although, *A. malaysiensis*, which is epidemic in Asian countries, may cause also human angiostrongyliasis [9]. In Thailand, *A. cantonensis*, *A. siamensis*, *A. malaysiensis*, and *Thaistrongylus harinasuti* have been recorded in several hosts [8,10]. *A. cantonensis* is the main causative agent of human angiostrongyliasis in Thailand, whereas *A. malaysiensis* is reported with increasing extent in the Greater Mekong area. To complete the life cycle, *A. cantonensis* and *A. malaysiensis* use snails and terrestrial slugs as intermediate hosts, and rodents as their final hosts [11]. Humans are an accidental host, and become infected by ingesting *Angiostrongylus* larvae present in snails, slugs, paratenic hosts or on contaminated vegetables [12–14]. Clinical manifestations of human angiostrongyliasis include severe headache, and neck stiffness with eosinophilic meningitis or meningoencephalitis. Most cases of the disease are reported in Thailand, Taiwan, and southern China [1].

The giant African land snail *Achatina fulica* is an important intermediate host for *A. cantonensis* [15]. In the 19[th] Century, this land snail was dispersed by humans across the Indian Ocean from Africa to India, Sri Lanka, and Southeast Asia [16]. In Thailand, most snails were accidentally moved across geographic locations on agricultural products or transportation containers. *A. fulica* is abundant in tropical climates with warm, mild year-round temperatures and high humidity [17]. The snail causes damage to vegetables and other food crops [18,19]. The spreading of *A. fulica* was affected also by dispersal of the rat lungworm, particularly in the Pacific [20]. Not surprisingly, the giant African land snail has been listed among the 100 worst invasive species and is considered the most damaging land snail in the world [17]. However, only a few genetic studies of *A. fulica* have been reported to date and none of them in Thailand.

Genetic characterization is important for the identification of parasitic pathogens, as well as to control the spreading of disease. Sequencing and phylogenetic studies of *A. cantonensis* based on mitochondrial or ribosomal genes have been used to identify and study the evolution and distribution of this species. Several genes or nucleotide regions from *A. cantonensis* have been used in genetic studies so far: 66-kDa protein [21], ribosomal transcribed spacer (ITS) regions [22,23], small subunit (SSU) ribosomal RNA (18S rRNA) [22,24,25], cytochrome c oxidase subunit I (COI) [9,26], and cytochrome b (*cytb*) [27,28]. In comparison, only a few studies have tried to characterize *A. malaysiensis* in Thailand [9]. Therefore, to gain further knowledge about the genetic make-up of *Angiostrongylus* spp. and its natural intermediate host, *A. fulica*, in Thailand, the present study sought to observe larval infection by *Angiostrongylus* in *A. fulica*. Analysis of *A. cantonensis* COI, SSU rRNA, ITS2, and *cytb* gene sequences and the ITS2 region of *A. malaysiensis* enabled construction of a phylogenetic tree. Finally, sequencing of the *A. fulica* COI gene, allowed for haplotype analysis and genetic structure characterization of this snail species.

## Methods

### Ethics statement

The experimental protocol for the use of animals (snail intermediate host) in this study was approved by the Center for Animal Research of Naresuan University (Project Ethics Approval No: NU-AQ610711). The biosafety protocol was approved by the Naresuan University Institutional Biosafety Committee (Project Approval No: NUIBC MI 61-08-50).

### Collection of *Achatina fulica*

*Achatina fulica* was randomly collected between May 2017 and July 2018 from 22 provinces across Thailand (Uttaradit, Chiang Rai, Chiang Mai, Nan, Phrae, Phitsanulok, Phetchabun, Bangkok, Lop Buri, Phra Nakhon Si Ayutthaya, Nakhon Sawan, Rayong, Nakhon Ratchasima, Buri Ram, Maha Sarakham, Chaiyaphum, Udon Thani, Nakhon Phanom, Prachuap Khiri Khan, Pattani, Chumphon, and Surat Thani) (**Fig 1** and **Table 1**). No specific permission was required for sampling snail in public locations. The snails were collected from several habitats (e.g., under leaf litter and under or above dried trees) by hand picking and were placed in a plastic box with air ventilation. The snails were then transported at ambient temperature to the Department of Microbiology and Parasitology, Faculty of Medical Science, Naresuan University, Phitsanulok, Thailand. All specimens were identified through comparison of shell morphology according to previous studies [29,30] Schotman (1989) and Jena et al. (2017); the conical shell of *A. fulica* was identified as wider at its operculum and tapering at its apex. The size of the shell was approximately 9.0–12.0 cm in length and 4.0–5.0 cm in width. The coloration and vertical stripes were dark brownish, and alternated by a cream tinge.

### Isolation of *Angiostrongylus* larvae from *Achatina fulica*

The body of *A. fulica* was removed from the shell. The mantle and foot of the land snail (approximately 25 mm$^3$) were cut by a razor blade and kept at -20˚C for further molecular analysis. To isolate *Angiostrongylus* larvae, most of the remaining snail's body was mixed with 50–100 mL of 0.7% pepsin solution (Acros Organics, Geel, Belgium) and minced in a blender. The mixed solution was transferred to a beaker and incubated in a water bath at 37˚C until the majority of the tissue was dissolved (1–2 h). The digested tissue solution was then placed in a Baermann apparatus, which consisted of a grass funnel connected to a short piece of rubber tubing at the outlet. The funnel, supported by a wire mesh, was covered with a layer of gauze and was let stand for 30–60 min to allow larval migration to the funnel's neck. The filtered liquid containing mainly larvae was transferred to a Petri dish. The larvae were identified as described previously [11]. *Angiostrongylus* larvae were collected using a sterile Pasteur pipette under a stereomicroscope, transferred to a 1.5-mL microcentrifuge tube, and stored at -20˚C for molecular analysis.

### Extraction of genomic DNA

Genomic DNA from individual land snails and from third-stage larvae of *Angiostrongylus* was extracted using the NucleoSpin® Tissue kit (Macherey-Nagel, Duren, Germany) following the manufacturer's instructions. The genomic DNA solution was checked by running it on a 0.8% agarose gel in 1× TBE buffer at 100 V. The gel was stained with ethidium bromide, followed by destaining with distilled water and photographed under u.v. light. The DNA solution was kept at -20˚C for further use.

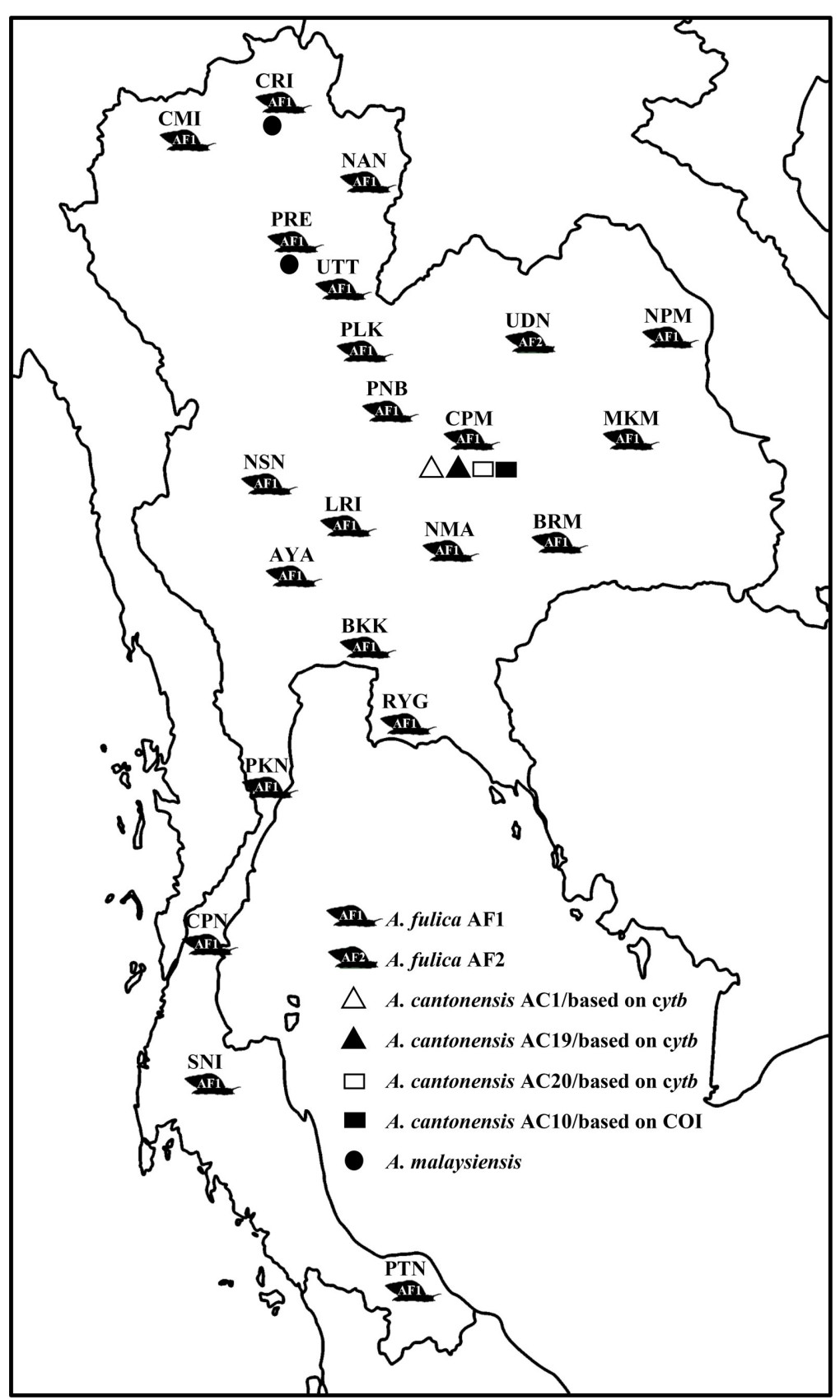

**Fig 1. Map of sampling location and distribution of *Achatina fulica*, *Angiostrongylus cantonensis*, and *A. malaysiensis* in Thailand.** Details of collection sites are given in Table 1.

## PCR and sequencing

PCR was used to amplify a partial region of the COI gene of *A. fulica*. A set of primers, AfCOI_F (5′–TGTGGGTTAGTTGGCACAGG–3′) and AfCOI_R (5′–TTAAGGGCGGGTACAC AGTC–3′), was designed based on the deposited GenBank sequence (accession no. KT290318) using the Primer-BLAST program. The primers were used to amplify a 319-bp fragment. The reaction mixture was prepared in a total volume of 30 μL containing 3 μL of 10× buffer (1×), 2.1 μL of 25 mM MgCl$_2$ (1.75 mM), 0.6 μL of 200 mM dNTPs (4 mM), 1.2 μL of 5 μM of each primer (0.2 μM), 0.3 μL of 5 U/mL Taq DNA polymerase (0.05 U/mL), 18.6 μL of distilled water, and 3 μL of DNA template (20–200 ng). The PCR conditions included initial denaturation at 95°C for 1 min, followed by 35 cycles of denaturation at 95°C for 1 min, annealing at 50°C for 40 s, extension at 72°C for 1 min, and final extension at 72°C for 5 min. The PCR was performed in a Biometra TOne Thermal cycler (Analytik Jena AG, Jena, Germany). The amplified products were analyzed by 1.2% agarose gel-electrophoresis, stained with ethidium bromide, destained with distilled water, and visualized and photographed under u.v. light. The amplified PCR products were then purified using a NucleoSpin® Gel and PCR Clean-up kit (Macherey-Nagel). Two volumes (56 μL) of buffer NTI were added to the tube containing the PCR product solution (28 μL). The mixture was then transferred onto a NucleoSpin® Gel and PCR Clean-up Column and centrifuged at 11,000 × *g* for 30 s. Buffer NT3 (700 μL) was added to the tube and this was centrifuged again at 11,000 × *g* for 30 s. The column was transferred to

**Table 1. Locations in Thailand where *Achatina fulica* was collected and number of corresponding COI haplotypes.**

| Regions | Location | Code | Latitude | Longitude | Number of samples | Haplotype name | Number of haplotypes |
|---|---|---|---|---|---|---|---|
| North | Uttaradit | UTT | 17.620088 | 100.099294 | 2 | AF1 | 2 |
| | Chiang Rai | CRI | 19.907165 | 99.830955 | 4 | AF1 | 4 |
| | Chiang Mai | CMI | 18.706064 | 98.981716 | 2 | AF1 | 2 |
| | Nan | NAN | 18.775631 | 100.773041 | 4 | AF1 | 4 |
| | Phrae | PRE | 18.144577 | 100.140283 | 4 | AF1 | 4 |
| Central | Phitsanulok | PLK | 17.036385 | 100.583513 | 3 | AF1 | 3 |
| | Phetchabun | PNB | 16.301669 | 101.119280 | 4 | AF1 | 4 |
| | Bangkok | BKK | 13.756330 | 100.501765 | 4 | AF1 | 4 |
| | Lop Buri | LRI | 14.799508 | 100.653370 | 1 | AF1 | 1 |
| | Phra Nakhon Si Ayutthaya | AYA | 14.353212 | 100.568959 | 3 | AF1 | 3 |
| | Nakhon Sawan | NSN | 15.693007 | 100.122559 | 4 | AF1 | 4 |
| East | Rayong | RYG | 12.707434 | 101.147351 | 3 | AF1 | 3 |
| Northeast | Nakhon Ratchasima | NMA | 14.979899 | 102.097769 | 2 | AF1 | 2 |
| | Buri Ram | BRM | 14.993001 | 103.102919 | 3 | AF1 | 3 |
| | Maha Sarakham | MKM | 16.013201 | 103.161516 | 4 | AF1 | 4 |
| | Chaiyaphum | CPM | 15.806817 | 102.031502 | 4 | AF1 | 4 |
| | Udon Thani | UDN | 17.413841 | 102.787232 | 4 | AF2 | 4 |
| | Nakhon Phanom | NPM | 17.392039 | 104.769550 | 3 | AF1 | 3 |
| South | Pattani | PTN | 6.761830 | 101.323254 | 4 | AF1 | 4 |
| | Chumphon | CPN | 10.493049 | 99.180019 | 4 | AF1 | 4 |
| | Surat Thani | SNI | 9.138238 | 99.321748 | 4 | AF1 | 4 |
| West | Prachuap Khiri Khan | PKN | 11.812367 | 99.797327 | 4 | AF1 | 4 |

**Table 2. Primers used for amplifying nucleotide regions in *Angiostrongylus*.**

| Gene or region | Primer/(Reference) | Approximate amplicon size (bp) | Target organism |
|---|---|---|---|
| SSU rRNA | Angi18S-1_forward<br>5′- AAAGTTAAGCCATGCATG -3′<br>Angi18S-2_reverse<br>5′- CATTCTTGGCAAATGCTTTCG -3′<br>[31] | 885 | *A. cantonensis* |
| *cytb* | Cytb-F<br>5′-TGAATAGACAGAATTTTAAGAG-3′<br>Cytb-R<br>5′-ATCAACTTAACATTACAGAAAC-3′<br>[27] | 853 | *A. cantonensis* |
| COI | AngiCOI_forward<br>5′- TTTTTTGGGCATCCTGAGGTTTAT -3′<br>AngiCOI_reverse<br>5′- CGAGGATAACCATGTAAACCAGC -3′ [9] | 605 | *A. cantonensis* |
| ITS2 | AngiITS2_forward<br>5'—ACATCTGGTTCAGGGTTGTT—3'<br>AngiITS2_ reverse<br>5'—AGCATACAAGCACATGATCAC—3'<br>[9] | 395 | *A. cantonensis*<br>*A. malaysiensis* |

a new 1.5-mL microcentrifuge tube, followed by addition of 30 μL of elution buffer and incubation at room temperature (25°C) for 1 min. The tube was centrifuged at 11,000 × *g* for 1 min. The purified PCR product was checked on a 1.2% agarose gel in 1× TBE buffer at 100 V. The gel was stained with ethidium bromide, followed by destaining with distilled water and photographed under u.v. light. The purified PCR products were shipped to Macrogen Inc., Seoul, Korea, for sequencing in both forward and reverse directions.

For *Angiostrongylus*, PCR was used to amplify the selected nucleotide regions (COI, ITS2, SSU rRNA, and *cytb* for *A. cantonensis* and ITS2 for *A. malaysiensis*). Specific primers are listed in **Table 2**. The reaction mixture was prepared in a total volume of 30 μL, containing 15 μL of EconoTaq® PLUS 2× Master mix (1×; Lucigen Corporation, Middleton, WI, USA), 1.5 μL of 5 μM of each primer (0.25 μM), 9 μL of distilled water, and 3 μL of DNA template (20–200 ng). PCR conditions for amplifying COI, ITS, and SSU rRNA were as described by Rodpai et al. (2016); whereas those for amplifying *cytb* were as described by Dusitsittipon et al. (2015). All PCR amplifications were conducted in a Biometra TOne Thermal cycler. The amplified products were analyzed by 1.2% agarose gel-electrophoresis as mentioned above, and then purified using a NucleoSpin® Gel and PCR Clean-up kit as mentioned above. The PCR products were sequenced in both the forward and reverse direction by Macrogen Inc.

## Sequence and phylogenetic analysis

All sequences were edited by viewing the peak of the chromatogram in SeqMan II software (DNASTAR, Madison, WI, USA). Species identification of *Angiostrongylus* and *Achatina* was confirmed by a BLASTN search, whereby similarity with known sequences in the NCBI database (http://blast.ncbi.nlm.nih.gov/Blast.cgi) was determined. The maximum likelihood (ML), neighbour joining (NJ) and maximum parsimony (MP) phylogenetic trees were constructed based on a Kimura 2-parameter model for SSU rRNA and *cytb* alignment, Tamura 3-parameter model for ITS2 alignment, and the General Time Reversible model for COI alignment with 1,000 bootstrap replicates using MEGA 7 software [32]. Bayesian analysis were performed based on Markov chain Monte Carlo method in MrBayes v3.2 [33–35]

Haplotype diversity (*h*) and nucleotide diversity (*π*) for *A. fulica* were calculated in ARLE-QUIN, version 3.5.1.2 [36]. Population pairwise $F_{ST}$ (p < 0.05) calculated in ARLEQUIN was used to infer the genetic structure of *A. fulica* from the six different sampled regions.

## Results

### Infection of *Angiostrongylus* in *Achatina fulica*

A total of 1,595 *Angiostrongylus* larvae were isolated from 343 *A. fulica* collected in 22 provinces across Thailand. These included 13 and 1,269 *A. malaysiensis* larvae isolated from *A. fulica* collected in Phrae and Chiang Rai provinces in northern Thailand, respectively; as well as 313 *A. cantonensis* larvae from *A. fulica* collected in Chaiyaphum province, northeast Thailand (**S1 Table**). A total of 41 *Angiostrongylus* spp. specimens and 74 individual land snails were randomly selected for further genetic studies.

### Genetic characterization of *Achatina fulica*

A partial region of the COI gene from 74 individual snails collected in different locations was amplified by PCR and sequenced. PCR amplicons were 319 bp in length. Based on an edited stretch of 291 bp, all sequences showed high similarity (99–100%) with the known COI sequence of *A. fulica* (GenBank accession no. MF415341). All 74 sequences of *A. fulica* COI in the present study were deposited in the NCBI database (GenBank accession nos. MK858335-MK858408). The phylogenetic tree of the 74 COI sequences of *A. fulica* collected across Thailand revealed a monophyletic cluster with bootstrap and Bayesian posterior probability values (61/62/-/85%). It could be grouped together with *A. fulica* from the United Kingdom and the United States (**Fig 2**).

Two haplotypes were identified and named as AF1 and AF2 (**Table 1** and **S1 Fig**). All nucleotides in haplotypes AF1 and AF2 were the same except for the nucleotide at position 161, which corresponded to a "T" in AF1 and a "C" in AF2. Haplotype AF1 was found in the northern, central, eastern, northeastern, southern, and western regions of Thailand; whereas haplotype AF2 was found in Udon Thani province, northeast Thailand. **Table 3** shows the haplotype and nucleotide diversity of *A. fulica* COI sequences. In northeastern Thailand, haplotype and nucleotide diversity were found among 20 specimens. Population pairwise $F_{ST}$ analysis revealed no genetic differentiation among populations from northern, central, eastern, northeastern, southern, and western Thailand (**Table 4**).

### Genetic characterization of *Angiostrongylus*

Partial sequences of SSU rRNA, COI, *cytb*, and ITS2 regions from 41 *Angiostrongylus* specimens were determined by PCR and sequencing. BLASTN search results relative to these four nucleotide regions are shown in **Table 5**. Based on 839 bp of the SSU rRNA gene, 14 specimens (GenBank accession nos. MK858285-MK858298) of *Angiostrongylus* showed 100% identity with *A. cantonensis* (GenBank accession no. KU528682). The maximum likelihood tree derived from all sequences of SSU rRNA from the present study was grouped together with *A. cantonensis* from Thailand (GenBank accession nos. KU528687 and KU528682) and Japan (GenBank accession no. AY295804) (**Fig 3**). Phylogenetic tree showed well support values (98/97/95/100%). There was no difference in p-distance for intraspecific divergence within *A. cantonensis* (**S2 Table**).

Based on a partial COI sequence (478 bp) of *Angiostrongylus*, 16 sequences (GenBank accession nos. MK734431-MK734446) from this study showed 100% similarity with the known sequence of *A. cantonensis* (GenBank accession no. KU532147). The maximum likelihood tree

Af240RYG_TH
Af242RYG_TH
Af244RYG_TH
Af255AYA_TH
Af256AYA_TH
Af257AYA_TH
Af265NSN_TH
Af266NSN_TH
Af267NSN_TH
Af269NSN_TH
Af272NAN_TH
Af273NAN_TH
Af274 NAN_TH
Af279NAN_TH
Af227PRE_TH
Af144CPM_TH
Af202NPM_TH
Af203NPM_TH
Af204NPM_TH
Af213PKN_TH
Af214PKN_TH
Af215PKN_TH
Af216PKN_TH
Af217UDN_TH
63/65/72/91 Af218UDN_TH
Af221UDN_TH
Af222UDN_TH
Af230PRE_TH
Af232PRE_TH
Af233LRI_TH
Af278PRE_TH
Af104PTN_TH
Af110PTN_TH
Af112PTN_TH
Af115PTN_TH
Af118CRI_TH
Af120CRI_TH
61/62/-/85 Af123CRI_TH
Af127CRI_TH
Af138CMI_TH
Af139CMI_TH
Af147CPM_TH
Af149CPM_TH
Af150CPM_TH
Af181SNI_TH
Af183SNI_TH
Af184SNI_TH
Af185SNI_TH
Af18BRM_TH
Af19BRM_TH
Af1NMA_TH
Af20BRM_TH
Af21PNB_TH
Af22PNB_TH
MF415341 *Achatina fulica* UK
AY148556 *Achatina fulica* USA
Af23PNB_TH
Af24PNB_TH
Af26MKM_TH
Af27MKM_TH
Af28MKM_TH
Af2NMA_TH
Af32MKM_TH
Af3PLK_TH
Af43UTT_TH
Af44UTT_TH
Af47CPN_TH
Af48CPN_TH
Af4PLK_TH
Af50CPN_TH
Af51CPN_TH
Af5PLK_TH
Af74BKK_TH
Af81BKK_TH
Af84BKK_TH
Af85BKK_TH
KT290318 *Achatina fulica* NG
KT290317 *Achatina fulica* NG
KT290310 *Archachatina marginata*

0.02

**Fig 2. Maximum likelihood phylogenetic tree generated from 74 sequences of a partial COI sequence (291 bp) of *A. fulica* collected across Thailand.** Support values (ML bootstrap/NJ bootstrap/MP bootstrap/Bayesian posterior probabilities) show above the branches of the phylogenetic tree. At the branches of the tree, a dash (-) instead of a numerical support value indicates that a certain grouping was not seen by that method of analysis. Bold letters indicate sequences obtained in the present study. *Archachatina marginata* was used as the out-group. TH, Thailand; UK, United Kingdom; USA, United States of America; NG, Nigeria.

of 16 *Angiostrongylus* COI sequences was grouped together with the identified haplotypes of *A. cantonensis* AC10 (GenBank accession no. KU532147) (Fig 4). All sequences were formed a monophyletic group with support values of 66/64/83/85%. There was no difference in p-distance for intraspecific divergence within *A. cantonensis* (S3 Table).

All ten cytb sequences (853 bp) from the present study (GenBank accession nos. MK858275-MK858284) showed 99–100% identity to the known sequence of *A. cantonensis* (GenBank accession no. KP721446). The phylogenetic tree of the cytb sequence revealed that ten sequences from the present study were closely related to *A. cantonensis* AC1 (GenBank accession no. KP721446), AC2 (GenBank accession no. KC995263), AC3 (GenBank accession no. KP721449), AC4 (GenBank accession no. KC995278), AC5 (GenBank accession no. KP721447), AC6 (GenBank accession no. KP721442), AC7 (GenBank accession no. KC995190), and AC8 (GenBank accession no. KC995205) (Fig 5). Six sequences from this study were similar to *A. cantonensis* haplotype AC1. In addition, three sequences were identified as belonging to the new haplotype AC19 with support values of 65/63/-/93% and one to the haplotype AC20 with support values of 62/78/56/-%. Comparison of nucleotide sequences between these two new haplotypes and 18 previously reported haplotypes is presented in S4 Table. Intraspecific distances within *A. cantonensis* ranged from <0.1% to 0.9% (S5 Table).

The ITS2 sequences (278 bp) from 20 specimens in the present study (GenBank accession nos. MK858299-MK858318) displayed 100% similarity to *A. cantonensis* (GenBank accession no. KU528692). A maximum likelihood tree showed that all 20 sequences fell in the *A. cantonensis* groups from Thailand (GenBank accession nos. KU528688 and KU528692), the Philippines (GenBank accession nos. EU636007 and EU636008), China (GenBank accession nos. HQ540546, HQ540549, and HQ540551), United States (GenBank accession no. KU528689), and Spain (GenBank accession no. GQ181112) (Fig 6). In addition, sixteen ITS2 sequences (268 bp) of *Angiostrongylus* larvae from this study (GenBank accession nos. MK858319-MK858334) showed 100% identity to *A. malaysiensis* (GenBank accession no. KU528697). Based on the maximum likelihood tree, all *A. malaysiensis* sequences fell in the *A. malaysiensis* groups from Thailand (GenBank accession no. KU528695), Laos (GenBank accession no. KU528697), and Myanmar (GenBank accession no. KU528694) (Fig 6). Two major clades of phylogenetic tree based on ITS2 sequences showed clearly distinguished between *A. cantonensis* and *A. malaysiensis* with support values of 100/97/100/100% and 100/

**Table 3. Haplotype diversity (h) and nucleotide diversity (π) for six populations of *A. fulica* in Thailand based on mitochondrial COI gene sequences.**

| Location | Number of samples | Haplotype diversity (*h*), mean±SD | Nucleotide diversity (*π*), mean±SD |
|---|---|---|---|
| Northern | 16 | 0 | 0 |
| Central | 19 | 0 | 0 |
| Eastern | 3 | 0 | 0 |
| Northeastern | 20 | 0.3368±0.1098 | 0.0011±0.0013 |
| Southern | 12 | 0 | 0 |
| Western | 4 | 0 | 0 |
| Total | 74 | 0.1037±0.0469 | 0.0003±0.0006 |

**Table 4. Population pairwise $F_{ST}$ from six populations of *A. fulica* based on mitochondrial COI gene sequences.**

| Populations | Northern | Central | Eastern | Northeastern | Southern | Western |
|---|---|---|---|---|---|---|
| Northern | 0.000 | | | | | |
| Central | 0.000 | 0.000 | | | | |
| Eastern | 0.000 | 0.000 | 0.000 | | | |
| Northeastern | 0.135 | 0.152 | -0.064 | 0.000 | | |
| Southern | 0.000 | 0.000 | 0.000 | 0.107 | 0.000 | |
| Western | 0.000 | 0.000 | 0.000 | -0.012 | 0.000 | 0.000 |

83/100/66%, respectively (**Fig 6**). Interspecific distances between *A. cantonensis* and *A. malaysiensis* sequences ranged from 14.8% to 15.5% (**S6 Table**). Intraspecific distances among *A. cantonensis* samples were <0.4–0.7%. There was no difference in p-distance for intraspecific divergence within *A. malaysiensis* (**S7 Table** and **S8 Table**).

## Discussion

The present study describes infection of *A. fulica* with *A. cantonensis* and *A. malaysiensis* in Thailand. Earlier studies on the prevalence of *A. cantonensis* reported infection rates of up to 90% [37,38]; these estimates were lowered in later studies to 36.4% [39]. Recently, several surveys of *A. cantonensis* larvae in *A. fulica* were reported from various provinces in Thailand, their infection rates ranging between 1.1% and 7.6% [26,27,40]. Such low infection rates may be ascribed to variability in the presence and abundance of *A. cantonensis* in different environments, as well as to abiotic factors, such as humidity and temperature. The distribution of infected rats, the species of rats, and the interactions between gastropods and rats may determine the prevalence of *A. cantonensis* in snails [41]. In addition, we found *A. fulica* infected with *A. malaysiensis* in the north (Phrae and Chiang Rai provinces) of the country; in agreement with a previous report that detected *A. malaysiensis* in the northern Mae Hong Sorn and Tak provinces. Because this nematode is found also in the south (Phatthalung and Phang Nga provinces) of the country, close to Malaysia [9,28], *A. fulica* is a possible intermediate host for *A. cantonensis* and *A. malaysiensis* throughout Thailand. Moreover, *A. fulica* is implicated in an increased distribution of *A. cantonensis* in China and Japan [29,42]. At present, most human angiostrongyliasis cases are reported in the northeast of Thailand, but transmission of *Angiostrongylus* species reflects the dispersal of intermediate and definitive hosts. Accordingly, spreading of the African giant land snail can potentially augment the dispersion of angiostrongyliasis cases.

In the present study, genetic characterization of *A. fulica* collected from across Thailand was studied based on sequencing of the COI gene. The phylogenetic tree showed a monophyletic group for *A. fulica* in Thailand, suggesting that a single lineage of this snail had been introduced in the country. Interestingly, this lineage was closely related to *A. fulica* from the United Kingdom and the United States. Accordingly, these giant African land snail populations may share a common ancestor that was brought to each country by human intervention. In a previous study, *A. fulica* from Odisha state in India was closely related to *A. fulica* from Bangalore, Kerala, Africa, Cameroon, and China [30]. That result differs from the present one relating the Thai, UK, and USA isolates to a common origin, and demonstrates the existence of multiple lineages of this snail around the world, all originating from the African continent. The population genetic structure of *A. fulica* in Thailand revealed no difference between the six sampled regions. This uniformity may be due to gene flow within the *A. fulica* population in Thailand. Indeed, *A. fulica* was first introduced into Thailand from Malaysia in 1937 [43].

**Table 5. BLASTN search based on SSU rRNA, ITS2, COI, and *cytb* regions of *Angiostrongylus* spp. (41 specimens) in Thailand.**

| Code | Host/Location | No. of larvae | BLASTN identity (%) | | | | Species identification |
|---|---|---|---|---|---|---|---|
| | | | COI | SSU rRNA | *cytb* | ITS2 | |
| AngC30CPM_TH | *A. fulica*/Chaiyaphum | 1 | 100 | ND | ND | ND | *A. cantonensis* |
| AngC31CPM_TH | *A. fulica*/Chaiyaphum | 1 | 100 | 100 | 100 | 100 | *A. cantonensis* |
| AngC32CPM_TH | *A. fulica*/Chaiyaphum | 1 | 100 | ND | ND | ND | *A. cantonensis* |
| AngC33CPM_TH | *A. fulica*/Chaiyaphum | 1 | 100 | ND | ND | ND | *A. cantonensis* |
| AngC35CPM_TH | *A. fulica*/Chaiyaphum | 1 | 100 | ND | 100 | 99 | *A. cantonensis* |
| AngC45CPM_TH | *A. fulica*/Chaiyaphum | 1 | 100 | 100 | ND | 100 | *A. cantonensis* |
| AngC46CPM_TH | *A. fulica*/Chaiyaphum | 1 | 100 | ND | ND | 100 | *A. cantonensis* |
| AngC47CPM_TH | *A. fulica*/Chaiyaphum | 1 | 100 | 100 | ND | 100 | *A. cantonensis* |
| AngC50CPM_TH | *A. fulica*/Chaiyaphum | 5 | 100 | ND | ND | 100 | *A. cantonensis* |
| AngC51CPM_TH | *A. fulica*/Chaiyaphum | 10 | ND | ND | 99 | 100 | *A. cantonensis* |
| AngC52CPM_TH | *A. fulica*/Chaiyaphum | 10 | 100 | 100 | 99 | ND | *A. cantonensis* |
| AngC53CPM_TH | *A. fulica*/Chaiyaphum | 10 | 100 | 100 | 100 | 100 | *A. cantonensis* |
| AngC72CPM_TH | *A. fulica*/Chaiyaphum | 10 | 100 | 100 | 99 | 100 | *A. cantonensis* |
| AngC73CPM_TH | *A. fulica*/Chaiyaphum | 10 | 100 | 100 | 100 | 100 | *A. cantonensis* |
| AngC74CPM_TH | *A. fulica*/Chaiyaphum | 10 | ND | ND | ND | 100 | *A. cantonensis* |
| AngC75CPM_TH | *A. fulica*/Chaiyaphum | 10 | ND | 100 | ND | 100 | *A. cantonensis* |
| AngC76CPM_TH | *A. fulica*/Chaiyaphum | 9 | 100 | 100 | 100 | 100 | *A. cantonensis* |
| AngC77CPM_TH | *A. fulica*/Chaiyaphum | 10 | 100 | 100 | ND | 100 | *A. cantonensis* |
| AngC78CPM_TH | *A. fulica*/Chaiyaphum | 10 | ND | ND | ND | 100 | *A. cantonensis* |
| AngC79CPM_TH | *A. fulica*/Chaiyaphum | 10 | ND | ND | ND | 100 | *A. cantonensis* |
| AngC80CPM_TH | *A. fulica*/Chaiyaphum | 8 | ND | 100 | 100 | 100 | *A. cantonensis* |
| AngC81CPM_TH | *A. fulica*/Chaiyaphum | 6 | ND | ND | ND | 100 | *A. cantonensis* |
| AngC83CPM_TH | *A. fulica*/Chaiyaphum | 10 | ND | 100 | ND | 100 | *A. cantonensis* |
| AngC84CPM_TH | *A. fulica*/Chaiyaphum | 10 | 100 | 100 | 99 | 100 | *A. cantonensis* |
| AngC85CPM_TH | *A. fulica*/Chaiyaphum | 10 | ND | 100 | ND | ND | *A. cantonensis* |
| AngM107CRI_TH | *A. fulica*/Chiang Rai | 10 | ND | ND | ND | 100 | *A. malaysiensis* |
| AngM15CRI_TH | *A. fulica*/Chiang Rai | 1 | ND | ND | ND | 100 | *A. malaysiensis* |
| AngM4CRI_TH | *A. fulica*/Chiang Rai | 1 | ND | ND | ND | 100 | *A. malaysiensis* |
| AngM14CRI_TH | *A. fulica*/Chiang Rai | 1 | ND | ND | ND | 100 | *A. malaysiensis* |
| AngM5CRI_TH | *A. fulica*/Chiang Rai | 1 | ND | ND | ND | 100 | *A. malaysiensis* |
| AngM6CRI_TH | *A. fulica*/Chiang Rai | 1 | ND | ND | ND | 100 | *A. malaysiensis* |
| AngM7CRI_TH | *A. fulica*/Chiang Rai | 1 | ND | ND | ND | 100 | *A. malaysiensis* |
| AngM21CRI_TH | *A. fulica*/Chiang Rai | 1 | ND | ND | ND | 100 | *A. malaysiensis* |
| AngM105CRI_TH | *A. fulica*/Chiang Rai | 10 | ND | ND | ND | 100 | *A. malaysiensis* |
| AngM70CRI_TH | *A. fulica*/Chiang Rai | 1 | ND | ND | ND | 100 | *A. malaysiensis* |
| AngM108CRI_TH | *A. fulica*/Chiang Rai | 1 | ND | ND | ND | 100 | *A. malaysiensis* |
| AngM109CRI_TH | *A. fulica*/Chiang Rai | 10 | ND | ND | ND | 100 | *A. malaysiensis* |
| AngM110CRI_TH | *A. fulica*/Chiang Rai | 10 | ND | ND | ND | 100 | *A. malaysiensis* |
| AngM40PRE_TH | *A. fulica*/Phrae | 1 | ND | ND | ND | 100 | *A. malaysiensis* |
| AngM26PRE_TH | *A. fulica*/Phrae | 1 | ND | ND | ND | 100 | *A. malaysiensis* |
| AngM39PRE_TH | *A. fulica*/Phrae | 1 | ND | ND | ND | 100 | *A. malaysiensis* |

ND, not determined

Five years after the first presumed entry, *A. fulica* population increased dramatically and expanded to several other parts of the country [44]. By analyzing the COI gene sequence, we

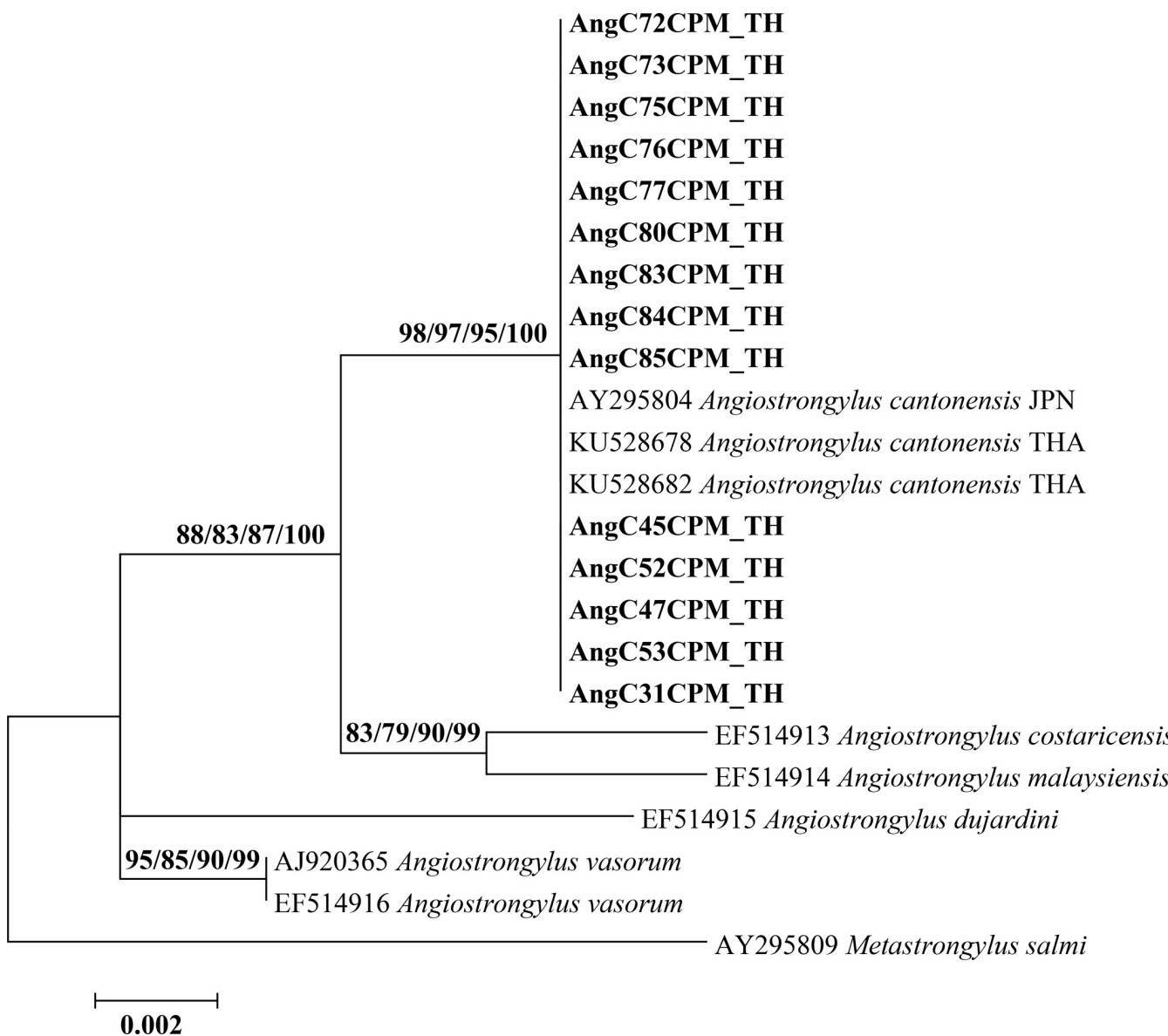

**Fig 3. Maximum likelihood phylogenetic tree of *A. cantonensis* based on a partial SSU rRNA sequence (839 bp).** Support values (ML bootstrap/NJ bootstrap/MP bootstrap/Bayesian posterior probabilities) show above the branches of the phylogenetic tree. Bold letters indicate sequences obtained in the present study. *Metastrongylus salmi* was used as the out-group. TH, THA, Thailand; JPN, Japan.

identified two haplotypes (AF1 and AF2) of *A. fulica*. However, diversity between the two haplotypes was detected only in the northeast of Thailand, possibly as a result of the founder effect [45]. Importantly, in the present study, *A. fulica* haplotype AF1 from the northeast and north of Thailand was naturally infected with *A. cantonensis* and *A. malaysiensis*; whereas haplotype AF2, which is restricted to the northeast of the country, was not infected with any of the two *Angiostrongylus* species. At present, it is difficult to explain why *A. fulica* haplotype AF1 seems to be more susceptible to *Angiostrongylus* infection than haplotype AF2.

Genetic characterization of *A. cantonensis* in the present study was achieved through sequencing of SSU rRNA, ITS2, COI, and *cytb* nucleotide regions. Based on the COI maximum likelihood tree, *A. cantonensis* (16 specimens) collected from Chaiyaphum province was

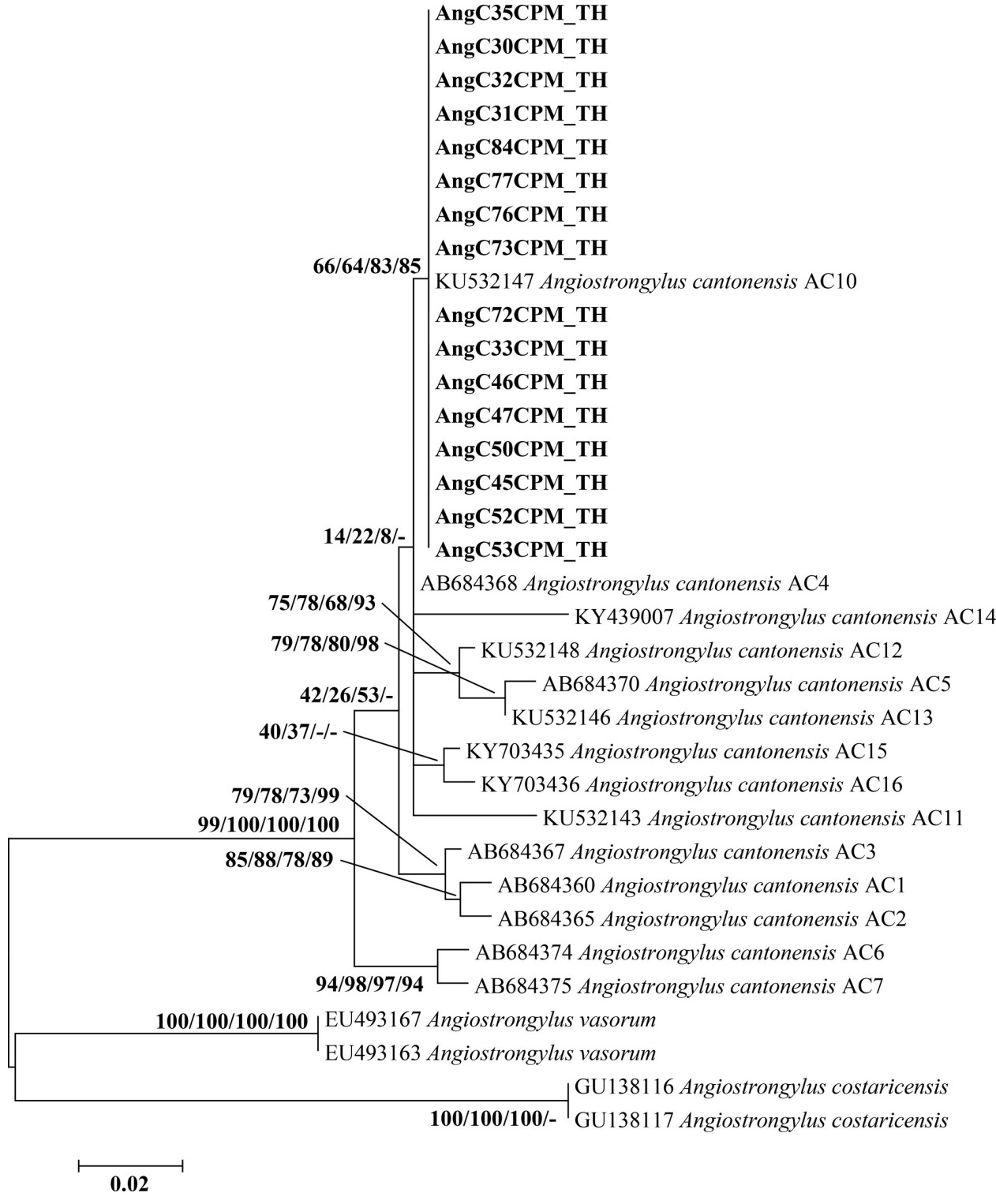

**Fig 4. Maximum likelihood tree based on a partial COI sequence (478 bp) of 16 samples of *A. cantonensis* from Thailand, together with *A. cantonensis* haplotypes AC1-AC16.** Support values (ML bootstrap/NJ bootstrap/MP bootstrap/Bayesian posterior probabilities) show above the branches of the phylogenetic tree. At the branches of the tree, a dash (-) instead of a numerical support value indicates that a certain grouping was not seen by that method of analysis. Bold letters indicate sequences obtained in the present study. *A. vasorum* and *A. costaricensis* were used as out-groups. TH, Thailand.

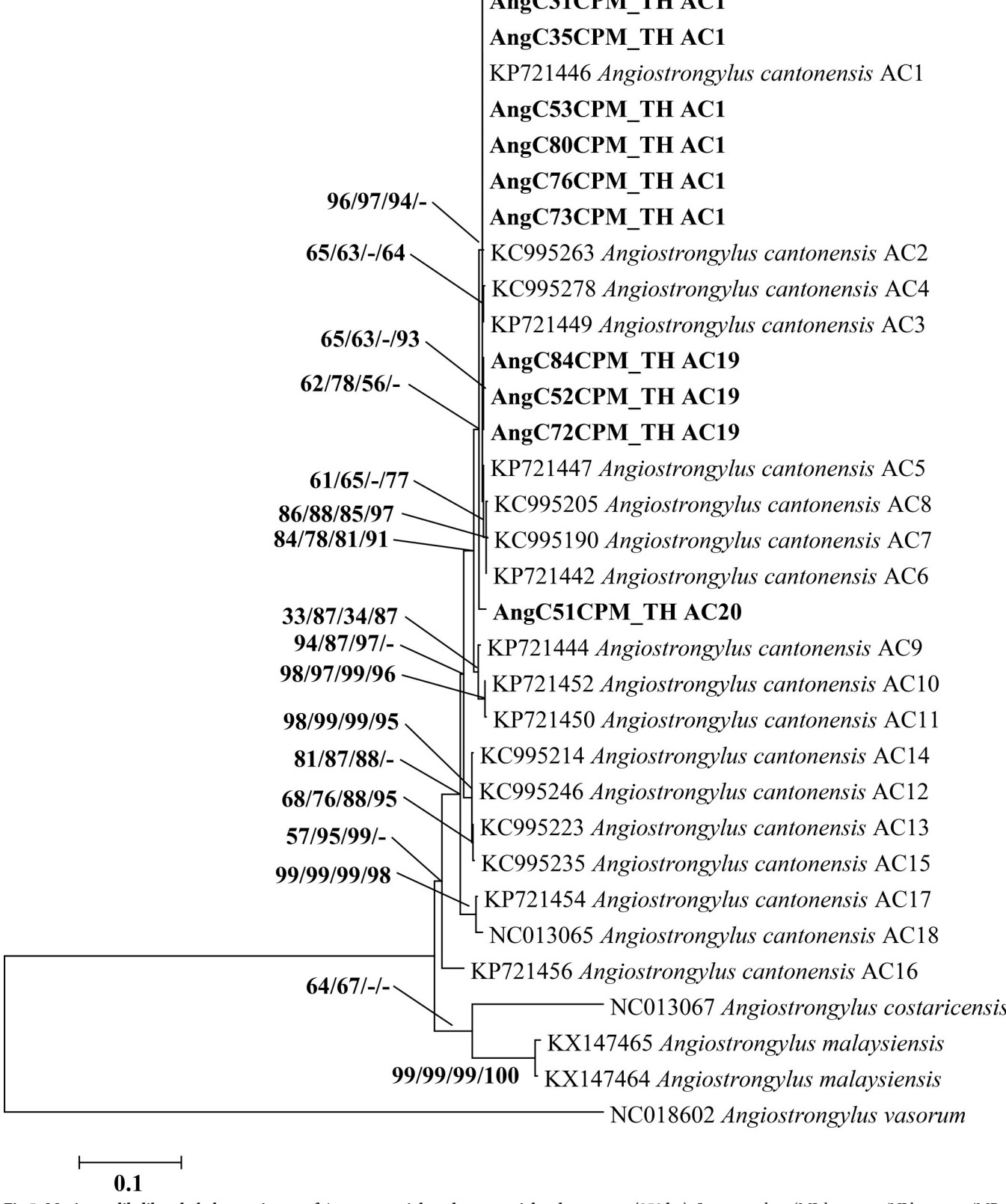

**Fig 5. Maximum likelihood phylogenetic tree of *A. cantonensis* based on a partial cytb sequence (853 bp).** Support values (ML bootstrap/NJ bootstrap/MP bootstrap/Bayesian posterior probabilities) show above the branches of the phylogenetic tree. At the branches of the tree, a dash (-) instead of a numerical

support value indicates that a certain grouping was not seen by that method of analysis. Bold letters indicate sequences obtained in the present study. *A. costaricensis*, *A. malaysiensis*, and *A. vasorum* are included in the tree. TH, Thailand.

closely related to *A. cantonensis* AC10, which was collected from the closely located Khon Kaen province. Sixteen COI-based haplotypes (AC1-AC16) of *A. cantonensis* have been reported from several parts of the world [9,31,46]. Haplotypes AC1, AC2, AC3, AC5, and AC7 were reported in Japan; haplotypes AC8 and AC9 were reported in Brazil; and haplotype AC6 was found in China (31, 46). In Thailand, the different haplotypes of *A. cantonensis* appeared confined to specific localities: haplotype AC4 to Bangkok in the central part of the country, AC10 and AC11 to Khon Kaen province in the northeast, AC13 to Surat Thani province in the south, AC14 to Kanchanaburi province in the west, AC15 to Trat province in the east, and AC16 to Chanthaburi province in the east. This distribution corresponds to our finding of the AC10 haplotype in Chaiyaphum province in the northeast of Thailand. Therefore, the COI region represents a good marker for studying the genetic evolution and differentiation of *Angiostrongylus* spp. [47], as well as to distinguish geographical isolates of *A. cantonensis* [46] and to identify its haplotypes [48].

In the present study, SSU rRNA sequences of *A. cantonensis* isolated from Chaiyaphum province shared a single phylogenetic group. All 14 sequence samples were closely related to the Thai and Japanese isolates. This was confirmed by the lack of difference between intraspecific distances within *A. cantonensis* isolates. Previous studies have reported little variation of the nuclear small subunit (SSU) rRNA sequences within a nematode species but substantial divergence among species, allowing for species differentiation [22,24,31]. Therefore, the SSU rRNA gene has been used to identify *A. cantonensis* and for the discrimination of *Angiostrongylus* species [24,31].

In this study, *A. cantonensis cytb* sequences (10 specimens) from Chaiyaphum province were closely related to AC1-AC8 *cytb* haplotypes found across several provinces in Thailand. However, most sequences (six specimens) were similar to *cytb* haplotype AC1, suggesting that this may be the dominant haplotype in Thailand. In addition, we identified two new *cytb* haplotypes: AC19 (three sequences) and AC20 (one sequence). Previous studies reported 15 haplotypes (AC1-AC15) based on the *cytb* sequence in Thailand; two haplotypes (AC16 and AC18) were reported in China; and one haplotype (AC17) was reported in Hawaii [27,28]. In Thailand, *cytb* haplotypes were distributed at random throughout the country; AC1 in Phitsanulok and Prachuap Khiri Khan provinces; AC2 in Prachuap Khiri Khan province; AC3 in Chiang Rai province; AC4 in Phitsanulok province; AC5 in Chiang Mai province; AC6 in Samut Prakan province and Bangkok; AC7 in Bangkok; AC8 in Kanchanaburi province; AC9 in Bangkok; AC10 in Nan, Surat Thani, and Nakhon Si Thammarat provinces; AC11 in Khon Kaen province; AC12 in Nan and Lop Buri provinces; AC13 in Maha Sarakham province; AC14 in Lop Buri province; and AC15 in Maha Sarakham province [28]. A larger sample size may reveal a clearer relationship between the *cytb* haplotype of this worm and localization in Thailand.

ITS2 sequences revealed differences between *A. cantonensis* and *A. malaysiensis*. The genetic distance between *A. cantonensis* and *A. malaysiensis* was 14.8–15.5%, whereas intraspecific distances among *A. cantonensis* were <0.4–0.7% and there was no intraspecific divergence within *A. malaysiensis*. Our findings are similar to those reported previously [9,23,49] and suggest that the ITS2 sequence might be useful for the identification of *Angiostrongylus* species [23,49].

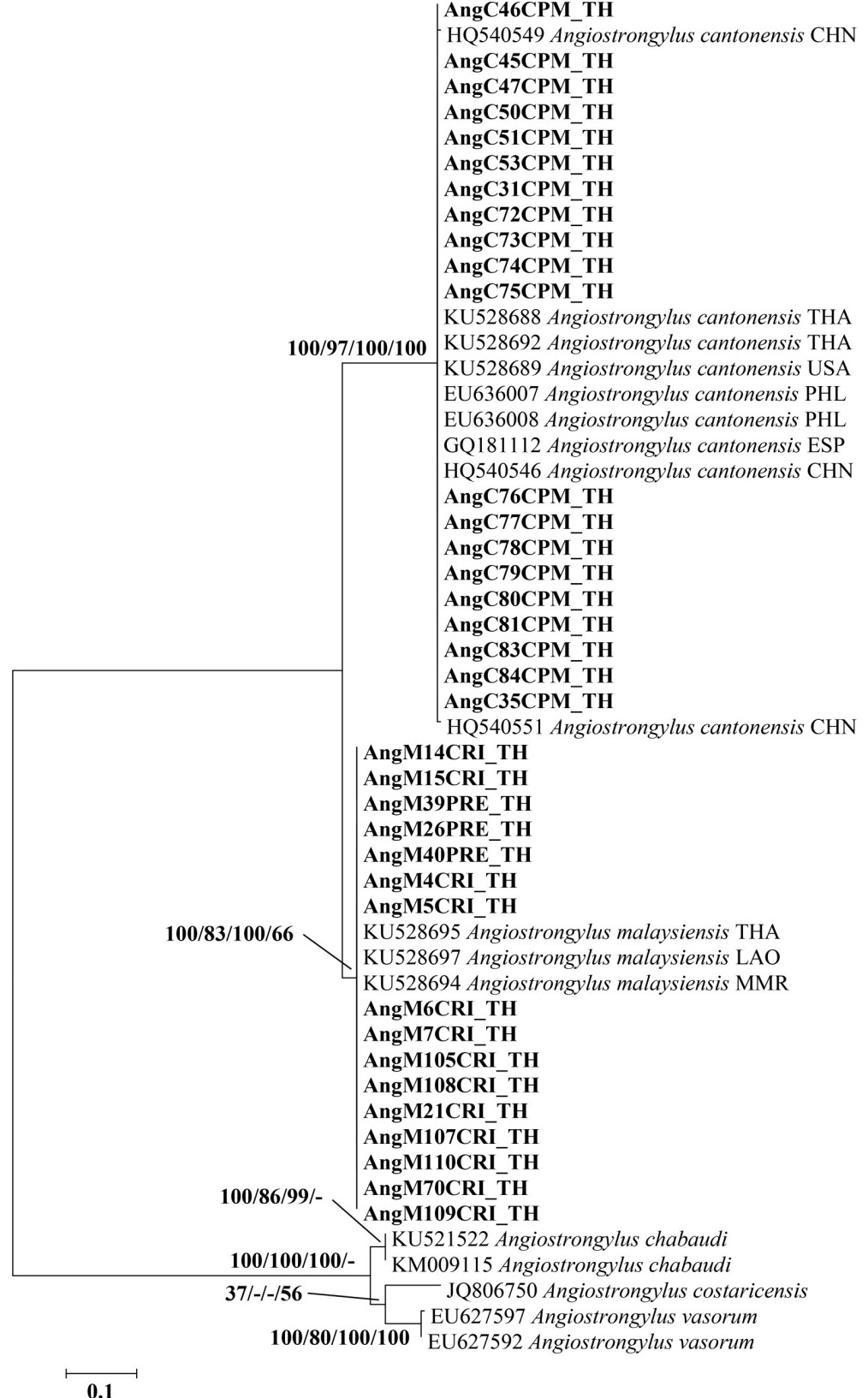

**Fig 6. Maximum likelihood phylogenetic tree of *A. cantonensis* and *A. malaysiensis* based on a partial ITS2 sequence (278 bp).** Support values (ML bootstrap/NJ bootstrap/MP bootstrap/Bayesian posterior probabilities) show above the branches of the phylogenetic tree. At the branches of the tree, a dash (-) instead of a numerical support value indicates that a certain grouping was not seen by that method of analysis. Bold letters indicate sequences obtained in the present study. *A. costaricensis*, *A. vasorum*, and *A. chabaudi* are included in the tree. TH, THA, Thailand; CHN, China; PHL, Philippines; LAO, Laos; MMR, Myanmar; USA, United States of America; ESP, España.

## Conclusions

In summary, we describe here the genetic characterization of *A. cantonensis* and *A. malaysiensis* isolated from the giant African snail *A. fulica* in Thailand. Two haplotypes (AF1 and AF2) of *A. fulica* were identified for the first time based on sequencing of the COI gene. Only haplotype AF1 of *A. fulica* was infected with *A. cantonensis* and *A. malaysiensis*. This confirmed that *A. cantonensis* and *A. malaysiensis* were found across the country. *A. fulica* is the main intermediate host for transmission of *Angiostrongylus* spp. in nature. The COI and *cytb* genes of *A. cantonensis* are suitable for phylogenetic studies, whereas the SSU rRNA gene is appropriate for identification. The ITS2 nucleotide region represents a good genetic marker for distinguishing between *A. cantonensis* and *A. malaysiensis*. Two new additional *cytb* haplotypes of *A. cantonensis* (AC19 and AC20) were identified in this study. A larger sample will help future studies on the genetics of this nematode species. This study provides basic genetic information about the parasite *Angiostrongylus* and its snail intermediate host, *A. fulica*.

## Supporting information

**S1 Fig. Mitochondrial DNA genealogy for 78 cytochrome c oxidase I (COI) sequences (74 sequences from Thailand and 4 sequences from other geographical regions) of *A. fulica* constructed by median joining network method.** Each haplotype is represented by a circle. Sizes of circles are relative to number of individuals sharing specific haplotype.
(TIF)

**S1 Table. Number of *Achatina fulica* used for artificial digestion method collected across Thailand.**
(PDF)

**S2 Table. Pairwise p-distances SSU rRNA gene within *A. cantonensis* samples.**
(XLS)

**S3 Table. Pairwise p-distances COI gene within *A. cantonensis* samples.**
(XLS)

**S4 Table. Variable nucleotide position within the *cytb* gene of *A. cantonensis*.**
(XLS)

**S5 Table. Pairwise p-distances *cytb* gene within *A. cantonensis* samples.**
(XLS)

**S6 Table. Pairwise p-distances ITS2 within *A. cantonensis* and *A. malaysiensis* samples.**
(XLS)

**S7 Table. Pairwise p-distances ITS2 within *A. cantonensis* samples.**
(XLS)

**S8 Table. Pairwise p-distances ITS2 within *A. malaysiensis* samples.**
(XLS)

## Acknowledgments

This work was supported by Naresuan University (Grant number R2562B079). We would like to thank Professor Dr. Pairot Pramual, Faculty of Science, Maha Sarakham University for his guidance in analyzing the population genetics of *Achatina fulica*.

## Author Contributions

**Conceptualization:** Abdulhakam Dumidae, Paron Dekumyoy, Aunchalee Thanwisai, Apichat Vitta.

**Data curation:** Abdulhakam Dumidae, Aunchalee Thanwisai, Apichat Vitta.

**Formal analysis:** Abdulhakam Dumidae, Paron Dekumyoy, Aunchalee Thanwisai, Apichat Vitta.

**Funding acquisition:** Apichat Vitta.

**Investigation:** Abdulhakam Dumidae, Chanakan Subkrasae, Apichat Vitta.

**Methodology:** Abdulhakam Dumidae, Pichamon Janthu, Chanakan Subkrasae, Aunchalee Thanwisai, Apichat Vitta.

**Project administration:** Abdulhakam Dumidae, Pichamon Janthu, Apichat Vitta.

**Resources:** Apichat Vitta.

**Software:** Aunchalee Thanwisai, Apichat Vitta.

**Supervision:** Apichat Vitta.

**Validation:** Apichat Vitta.

**Visualization:** Apichat Vitta.

**Writing – original draft:** Abdulhakam Dumidae, Pichamon Janthu, Chanakan Subkrasae, Paron Dekumyoy, Aunchalee Thanwisai, Apichat Vitta.

**Writing – review & editing:** Apichat Vitta.

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
