## [Decision Letter · Decision Letter 0]

22 Jul 2019

PONE-D-19-18473

Genetic characterization of Angiostrongylus larvae and their natural intermediate host, Achatina fulica, in Thailand

PLOS ONE

Dear Dr. Vitta,

Thank you for submitting your manuscript to PLOS ONE. After careful consideration, we feel that it has merit but does not fully meet PLOS ONE’s publication criteria as it currently stands. Therefore, we invite you to submit a revised version of the manuscript that addresses the points raised during the review process.

The MS brings informative results of the distribution of Angiostrongylus and the snail intermediate host in Thailand.

The comments and suggestions of the reviewers were complementary and the authors should address all the items suggested if possible.     

Improvement on the text is necessary. Organizing better the ideas, specially in the discussion. The methodology should be written in a more concise manner. The map presented as a supplementary material could be a figure in the article, with data of snails/haplotypes and Angiostrongylus/haplotypes. Also in the discussion, the correlation of haplotype and final hosts (human infection if possible) would be an interesting and informative point in this MS.

We would appreciate receiving your revised manuscript by Sep 05 2019 11:59PM. To enhance the reproducibility of your results, we recommend that if applicable you deposit your laboratory protocols in protocols.io, where a protocol can be assigned its own identifier (DOI) such that it can be cited independently in the future. For instructions see: http://journals.plos.org/plosone/s/submission-guidelines#loc-laboratory-protocols

We look forward to receiving your revised manuscript.

Kind regards,

Marcello Otake Sato, Ph.D., D.V.M.

Academic Editor

PLOS ONE

Journal Requirements:

Additional Editor Comments:

The MS brings informative results of the distribution of Angiostrongylus and the snail intermediate host in Thailand. However, improvement on the text is necessary. The methodology should be written in a more concise manner. The map presented as a supplementary material could be a figure with data of snails/haplotypes and Angiostrongylus/haplotypes. Also in the discussion, the correlation of haplotype and human infection (if possible) would be an interesting and informative point in this MS.

Reviewers' comments:

Reviewer's Responses to Questions

**Comments to the Author**

1. Is the manuscript technically sound, and do the data support the conclusions?

Reviewer #1: Yes

Reviewer #2: Yes

2. Has the statistical analysis been performed appropriately and rigorously? 

Reviewer #1: I Don't Know

Reviewer #2: N/A

3. Have the authors made all data underlying the findings in their manuscript fully available?

Reviewer #1: No

Reviewer #2: Yes

4. Is the manuscript presented in an intelligible fashion and written in standard English?

Reviewer #1: Yes

Reviewer #2: Yes

5. Review Comments to the Author

Reviewer #1: PONE-D-19-18473

This is a very nice set of data, with genetic studies of Angiostrongylus and its intermediate host in a wide an important endemic area.

The title: what do authors mean by “natural”? I can think of two meanings: “that can be found naturally infected” and “well-adapted”. I suggest to remove this word from the title since it does not add any fundamental information besides the main objective.

Line 73: As commented above, I see no reason to include “in nature”.

Line 84- Sure that genetic studies are fundamental, but the linkage between this study and “disease control” or specifically angiostrongyliasis control is not clear. Perhaps authors may re-write or clarify.

Line 100- “…an invasive land snail..” is not needed here as this was just addressed in the previous section.

Tables- formatting shall be revised according to PLoS One instructions.

Figures- They also need improvement for better quality.

Figure 1- the map of Thailand showing places where collection was performed. A much nicer presentation would be the indication of collection places with dots corresponding to exact geographic coordinates (geo-referencing). This shall be the informative mode of presenting geographic occurrence data, since just painting the geographic area of province is less informative.

Results- one very important information is the distribution of individual larvae counts. I strongly suggest authors to provide this information ina suppplementary file or to add it in manuscript and explore it. It gives a very nice opportunity to discuss the degree of parasite-host adaptation. It also enriches the contribution of this nice work.

Reviewer #2: Comment to MS: MS. No. PONE-D-19-18473 Genetic characterization of Angiostrongylus larvae and their natural intermediate host, Achatina fulica, in Thailand

This study revealed genetic diversity of Angiostrongylus larvae and Achatina fulica, natural intermediate host. New findings of A. fulica haplotypes were found (AF1 and AF2) based on sequencing of the COI gene. Two new additional cytb haplotypes of A. cantonensis (AC19 and AC20) were also identified in this study. Scientific sound is good. Manuscript is well written. This is useful for more understanding of system study to explore the epidemiology of Angiostrongylus spp. and Achatina fulica intermediate host.

Specific comment:

For ensure phylogenetic analysis, in addition to using the maximum-likelihood (ML) method in Phylogenetic analysis of Figures 1, 2 and 3, the authors should also use other methods such as (Bayesian analyses (BI) maximum-parsimony (MP) and neighbour-joining (NJ) analyses implemented as reported by as Ref 9: Rodpai R, Intapan PM, Thanchomnang T, Sanpool O, Sadaow L, Laymanivong S, et al. Angiostrongylus cantonensis and A. malaysiensis Broadly Overlap in Thailand, Lao PDR, Cambodia and Myanmar: A Molecular Survey of Larvae in Land Snails. PLoS One. 2016;11(8): e0161128. https://doi.org/10.1371/journal.pone.016112.

6. PLOS authors have the option to publish the peer review history of their article (what does this mean?). If published, this will include your full peer review and any attached files.

Reviewer #1: Yes: Carlos Graeff-Teixeira

Reviewer #2: No

---

## [Author Response · Author response to Decision Letter 0]

11 Aug 2019

We have changed the manuscript as reviewers suggested. Details are in the attached file.

---

## [Decision Letter · Decision Letter 1]

18 Sep 2019

Genetic characterization of Angiostrongylus larvae and their intermediate host, Achatina fulica, in Thailand

PONE-D-19-18473R1

Dear Dr. Vitta,

We are pleased to inform you that your manuscript has been judged scientifically suitable for publication and will be formally accepted for publication once it complies with all outstanding technical requirements.

With kind regards,

Marcello Otake Sato, Ph.D., D.V.M.

Academic Editor

PLOS ONE

Additional Editor Comments (optional):

Reviewers' comments:

Reviewer's Responses to Questions

**Comments to the Author**

1. If the authors have adequately addressed your comments raised in a previous round of review and you feel that this manuscript is now acceptable for publication, you may indicate that here to bypass the “Comments to the Author” section, enter your conflict of interest statement in the “Confidential to Editor” section, and submit your "Accept" recommendation.

Reviewer #1: All comments have been addressed

Reviewer #2: All comments have been addressed

2. Is the manuscript technically sound, and do the data support the conclusions?

Reviewer #1: Yes

Reviewer #2: Yes

3. Has the statistical analysis been performed appropriately and rigorously? 

Reviewer #1: I Don't Know

Reviewer #2: Yes

4. Have the authors made all data underlying the findings in their manuscript fully available?

Reviewer #1: Yes

Reviewer #2: Yes

5. Is the manuscript presented in an intelligible fashion and written in standard English?

Reviewer #1: Yes

Reviewer #2: Yes

6. Review Comments to the Author

Reviewer #1: Authors have adequately answered all questions and comments.

Reviewer #2: The revised manuscript have been done as standard of the journal and revised as my suggestions. I have no comment again.

7. PLOS authors have the option to publish the peer review history of their article (what does this mean?). If published, this will include your full peer review and any attached files.

Reviewer #1: Yes: Carlos Graeff-Teixeira

Reviewer #2: No

---

## [Editor Report · Acceptance letter]

20 Sep 2019

PONE-D-19-18473R1 

Genetic characterization of *Angiostrongylus* larvae and their intermediate host, *Achatina fulica*, in Thailand 

Dear Dr. Vitta:

I am pleased to inform you that your manuscript has been deemed suitable for publication in PLOS ONE. Congratulations! Your manuscript is now with our production department. 

With kind regards,

on behalf of

Dr. Marcello Otake Sato 

Academic Editor

PLOS ONE